# Gallic Acid as a Non-Selective Inhibitor of α/β-Hydrolase Fold Enzymes Involved in the Inflammatory Process: The Two Sides of the Same Coin

**DOI:** 10.3390/pharmaceutics14020368

**Published:** 2022-02-06

**Authors:** Marcos Hikari Toyama, Airam Rogero, Laila Lucyane Ferreira de Moraes, Gustavo Antônio Fernandes, Caroline Ramos da Cruz Costa, Mariana Novo Belchor, Agatha Manzi De Carli, Marcos Antônio de Oliveira

**Affiliations:** 1BIOMOLPEP Group, Biosciences Institute, São Paulo State University (UNESP), São Vicente 11330-900, São Paulo, Brazil; airam.roggero@hotmail.com (A.R.); laila_luciane@hotmail.com (L.L.F.d.M.); gustavoopfernandes2014@gmail.com (G.A.F.); carolbert@gmail.com (C.R.d.C.C.); belchor.mariana@gmail.com (M.N.B.); agatha.manzi@yahoo.com.br (A.M.D.C.); scaffix@gmail.com (M.A.d.O.); 2Postgraduate Program in Biotechnosciences, Federal University of ABC, Santo André 09210-580, São Paulo, Brazil

**Keywords:** gallic acid, edema, myotoxic effect, snake venom, phospholipase A2

## Abstract

(1) Background: Gallic acid (GA) has been characterized as an effective anti-inflammatory, antivenom, and promising drug for therapeutic use. (2/3) Methods and Results: GA was identified from ethanolic extract of fresh pitanga (*Eugenia uniflora*) leaves, which was identified using commercial GA. Commercial GA neutralized the enzymatic activity of secretory PLA2 (sPLA2) by inhibiting the active site and inducing changes in the secondary structure of the enzyme. Pharmacological edema assays showed that GA strongly decreased edema when the compound was previously incubated with sPLA2. However, prior treatment of GA (30 min before) significantly increased the edema and myotoxicity induced by sPLA2. The molecular docking results of GA with platelet-acetylhydrolase (PAF-AH) and acetylcholinesterase reveal that this compound was able to interact with the active site of both molecules, inhibiting the hydrolysis of platelet-activating factor (PAF) and acetylcholine (ACh). (4) Conclusion: GA has a great potential application; however, our results show that this compound can also induce adverse effects in previously treated animals. Additionally, the increased edema and myotoxicity observed experimentally in GA-treated animals may be due to the inhibition of PAF-AH and Acetylcholinesterase.

## 1. Introduction

Gallic acid (GA) is widely present in the plant kingdom and can be found in its free form in vegetal tissues, in the esters form, or as conjugated esterified molecules. The amount of this compound can vary depending on external stimuli, such as chemical and microbiological stressors, and is strongly associated with defense against predation by insects. In addition, some studies show that GA inhibits a range of esterases/lipases [1,2,3,4,5]. α/βhydrolase fold proteins are a major superfamily of enzymes that catalyze a wide range of reactions, featuring a conserved Ser-His-Acid catalytic triad. This family encompasses a variety of enzymes, including esterases, proteases, lipases, dehalogenases, peroxidases, and epoxide hydrolases. They all contain a catalytic triad composed of a nucleophile (typically serine, occasionally aspartate or cysteine), an anionic side chain, and an intervening histidine from the catalytic triad, located between the core and cap domains. This enzymatic fold is found throughout all domains of organisms, ranging from humans to plants, bacteria, and even viruses [6,7]. Moreover, this characteristic has been found in several important enzymes involved in inflammation, such as intracellular and extracellular Platelet-Activating Factor-Acetylhydrolase (PAF-AH). These enzymes are designated as PLA2 associated with lipoprotein (Lp-PLA2) and play a role in the special regulation of extracellular PAF concentration produced during inflammation, which is induced by the phospholipase A2 cascade that includes Arachidonic Acid (AA) production [8,9,10]. Other important enzymes that exhibit a similar protein α/β hydrolase fold are the Cholinesterase-like enzymes [11] and both types of cytosolic (cPLA2) and secretory phospholipase A2 (sPLA2) found in cells and snake venom, respectively [12]. sPLA2 is a type of α/β hydrolase folding enzyme, and this structural motif is also found in several enzymes which are involved in the inflammatory process resolution, such as PAF-AH and acetylcholinesterase [11,12].

Studies with natural compounds are a very seductive area that aims to find potential drugs that can be used to synthesize new molecules. Several articles in the literature suggest the beneficial effects of GA against phospholipase A2 activity. Costa et al., 2021 [13] showed that GA was able to inhibit and neutralize the pharmacological activity of BthTx-I (catalytically inactive PLA2). Furthermore, in all these models, the compounds were always preincubated with PLA2 isolated from venom [14,15,16,17], and, as we have confirmed, the preincubation leads to the complete neutralization of the enzymatic and pharmacological effects of this enzyme. Due to these beneficial properties and the apparent lack of toxicity of GA, this compound has been extensively marketed as a food supplement for daily use [13,18]. However, there is no study regarding the effect of GA inoculation on animals prior to sPLA2 administration and its effects on the animal body. Thus, the big question in this study was: when animals are previously treated with GA, could this compound neutralize the inflammatory and myotoxic action induced by sPLA2 from rattlesnake venom? Hence, in this short communication, we show that GA, when previously applied, increased the pharmacological effects induced by PLA2, and we reveal essential interactions of this compound with these crucial enzymes through molecular docking studies, an essential tool to understand in vitro and in vivo studies. Thus, we presented the other side of the coin: the experimental side effects of GA and the need for the compound to be used as a drug and not sold without a prescription or indication.

## 2. Materials and Methods

### 2.1. Venoms, Animals, and Reagents

*Crotalus durissus terrificus* (Cdt) venom was purchased from Bio-Agents Serpentarium (Batatais, São Paulo, Brazil). Analytical HPLC and sequencing-grade solutes and solvents were purchased from various suppliers (Bio-Rad, Hercules, CA, USA; Sigma-Aldrich, St. Louis, MO, USA; Boehringer Mannheim, São Paulo, SP, Brazil and Applied Biosystems; San Francisco, CA, USA). Female Swiss mice (~25 g) used in the pharmacological assays were obtained from the Multidisciplinary Center of Biological Investigations (CEMIB-UNICAMP). All animal experiments were approved by the Ethics Committee of State University of São Paulo/I.B./São Vicente under Protocol number: 11/2018-CEUA, and Protocol n° 10/2018-CEUA, both approved in 19 March 2019.

### 2.2. GA Identification

In this work, we used commercial GA to identify the prepared pitanga leaf samples based on the methods described by [15], including sample preparation using ultrasound-assisted extraction followed by a concentration. The material was cleaned using solid-phase extraction cartridges prior to fractionation on reversed-phase HPLC using an analytical C18 column (Sigma-Aldrich, St. Louis, MO, USA) (250 × 4.6 mm, 5 μm) and mobile phase water (A)/acetonitrile (B). GA identification was confirmed by commercial gallic acid marker (Gallic acid monohydrate, ACS 98%, Sigma-Aldrich, St. Louis, MO, USA) and the respective UV scanning of both compounds from sample and standard.

### 2.3. sPLA2 Purification, GA Incubation, and Circular Dichroism Spectroscopy

sPLA2 purification and its incubation with GA were performed using the method described by Cotrim et al.,2011 [16]. PLA2 purity was confirmed through HPLC and its enzymatic activity. Then, this enzyme was used for the pharmacological and biochemical assays. In the first step, sPLA2 was incubated with the protein, and the product of the sPLA2: GA incubation was purified to clear the unbound gallic acid complex. Samples of sPLA2: GA were also subjected to circular dichroism assays.

### 2.4. Enzymatic and Pharmacological Activity (Edema and Myonecrosis)

Enzymatic and pharmacological activities were performed following the method described by Toyama et al., 2014 [17]. sPLA2 activity and the GA potential to inhibit this enzyme were measured using a 96-well plate assay using 4-nitro-3-octanoyloxy-benzoic acid (NOBA or NOB, manufactured by BIOMOL, Plymouth Meeting, PA, USA) as the substrate. Enzyme activity, which was expressed as the initial velocity of the reaction (Vo), was calculated based on the increase in absorbance after 20 min. All assays were performed with a SpectraMax 340 multiwell plate reader (Molecular Devices, Sunnyvale, CA, USA) using the absorbance at 425 nm.

To confirm the GA potential to inhibit sPLA2 enzymatically, the paw edema assay was performed through the samples administration via right posterior sub plantar injection using randomly-chosen Swiss female mice (~25 g, *n* = 5). The first experiment included a previous incubation of sPLA2 with GA, a positive control, which was the protein with a saline solution, and the negative control (saline solution). Paw volumes were measured immediately before the injection and at selected time intervals (data not shown). In the second assay, we investigated the effect of the previous inoculation by intraperitoneal rout (i.p.) of GA dissolved in saline in a single dose of 50 µL per animal 30 min before sPLA2 injection in the paw. All volumes administered on animals’ paws were 20 µL, always following the same concentrations of the samples: 10 µg/per animal of sPLA2, 1 mM of GA and NaCl at 0.9%.

Myonecrosis was measured by liberation of creatine kinase (CK) from damaged muscle cells using the same animal model and treatments explained above. Hence, in the first myonecrosis assay, GA was previously incubated for 30 min with sPLA2, and then the samples were injected into the gastrocnemius muscle (data not shown). The second treatment was made through the previous GA administration in the peritoneum (50 µL) 30 min before the sPLA2 injection into the gastrocnemius muscle. Control groups (*n* = 5) were submitted to the same procedure; however, the negative control was inoculated with 20 µL of 0.9% NaCl, and 20 µL (1 mM µg) of the purified compound, and in the positive control, 20 µL (10 µg) of isolated sPLA2 was administered. The volumes applied into the muscle were 20 µL, and the concentrations of each sample were described in the previous paragraph. After 60 min, the animals’ blood was harvested from the tail in a heparinized tube, which was centrifuged and frozen. Seric creatine kinase (CK) levels were determined according to the manufacturer of the kit (Bioliquid, Pinhais, Brazil), and the data were expressed in units per liter (U/L).

### 2.5. Molecular Modeling (Docking) and Enzymatic Inhibition

The PDB (Protein Data Bank-https://www.rcsb.org accessed on 21 December 2021) and NCBI (NCBI-https://www.ncbi.nlm.nih.gov accessed on 21 December 2021) databases were used in this analysis to find the amino acid sequence of all proteins. Information on the structure of GA was taken from the PubChem platform (https://pubchem.ncbi.nlm.nih.gov accessed on 21 December 2021). The crystallographic model chosen as the best model for the construction of the theoretical structural model to PLA2 from Cdt was 2qog and Acetylcholinesterase (4m0e) and PAF-AH (3D59) from human models. The SWISS-MODEL platform (https://swissmodel.expasy.org accessed on 21 December 2021) and the Chimera 1.14 program (Ucsf Chimera, 2004) were used to assemble the structural molecular model of the protein and to evaluate the general possibilities of the proteins binding with GA. After a previous study of the mechanisms and the essential residues of these proteins, molecular anchoring experiments were performed by Autodock Vina. All molecules were prepared previously, adding polar hydrogen atoms and aggregating the Kollman charges. Then, the files were converted in PDBQT to perform the calculations of the energy maps (Grid Box). The size was chosen to enclose all amino acids from the catalytic site’s microenvironment. The results were taken using the tools Discovery Studio 4.0, LIGPLOT+, and Pymol v 2.4 to evaluate the binding energies, distances, and orientations of molecules in the microenvironment of the active site of PLA2. The Molinspiration platform was used to better visualize the structure of the compounds.

### 2.6. Statistical Analyses

Results were reported as means ± SD of replicate experiments. The significance of differences between means was assessed by an analysis of variance, followed by a Dunnett’s test when several experimental groups were compared to the control group. The confidence limit for significance was 5%.

## 3. Results

### Previous Injection of GA and Its Effects

Native sPLA_2_ exhibited a maximum edema value at 30 min, with a swelling value of 0.22 ± 0.013 mL (*n* = 5, and *p* < 0.05 *). GA treatment inoculated 30 min before sPLA2 administration, a condition called sPLA2:GA (1 mM/30’), showed edema of 0.26 ± 0.012 mL (*n* = 5, and *p* < 0.05 *) at 60 min of the edema experiment. Hence, the single dose of 50 microliters (1 mM) administered previously did not decrease edema in this treatment and revealed a similar profile of edema induced by sPLA2 with saline. However, in Figure 1A, it is possible to notice that the maximum edema peak induced by native sPLA2 was reached at 30 min, whereas in the animals with GA previously injected, edema reached a maximum at 60 min. Additionally, this compound promoted more persistent edema until the time of 240 min, besides being two times higher. Using the PAF kit (Mouse PAF ELISA Kit, EM 1261 Wuhan Fine Biotech Co., Ltd., Wuhan, Hubei, China), PAF concentration was evaluated in the plasma. The animals treated with saline and GA showed values of 0.13 ± 0.04 ng/mL (*n* = 5, and *p* < 0.05 *) and 0.15 ± 0.02 ng/mL (*n* = 5, and *p* <0. 05 *), respectively. Samples from animals treated with native PLA2 showed values of 0.21 ± 0.07 ng/mL (*n* = 5, and *p* < 0.05 *) and in animals previously treated with the compound after sPLA2 injection presented edema values of 0.78 ng/mL (*n* = 5, and *p* < 0.05 *).

Figure 1B reveals the myotoxic activity induced by native sPLA2 and sPLA2 injected in the muscle after GA previous treatment. The extent of damage caused by sPLA_2_ to skeletal muscles was obtained by quantifying the CK levels, which are widely used as an indirect marker of muscle damage. A total of 60 min after the native sPLA_2_ injection in the muscle, the CK value in animals previously treated with the compound (GA, 1 mM/30’) was 456 ± 46 U/L (*n* = 5, and *p* < 0.05 *), whereas for animals not exposed to any treatment (saline 30’), CK level found in plasma was 362 ± 27 U/L (*n* = 5, and *p* < 0.05 *). GA showed the same level found in animals under saline treatment. Thus, GA previously administered potentializes the myotoxic effect induced by sPLA2.

The sPLA2 incubated with GA previous to the administration on mice show an enzymatic, edematogenic, and myotoxic inhibition by the compound (data not shown), since edema reveals a volume of 0.09 ± 0.03 mL (*n* = 5, and *p* < 0.05 *) at the peak, and CK levels reached values of 167 ± 32 U/L (*n* = 5, and *p* < 0.05 *). Docking results reveal a strong molecular interaction of the compound with the enzyme active site, interacting with crucial amino acid residues located in the N-terminal region (F5), with Histidine (H48), Aspartic acid (D49). In Figure 2A, simulations performed with the Discovery program also suggest an interaction with the calcium ion. Also, results from the LigProt+ program suggest that GA hydroxyl groups (OH) interact with the side chain of Aspartic Acid (D49) and with the side chain of Cystein 45 (Cys45) besides the presence of several other interactions with Histidine 48 (H48) and with N-terminal Phenylalanine (F5) (Figure 2B). Thus, in Figure 2C, we present a possible interaction site of GA with sPLA2, exhibiting the involved residues that are crucial for the enzymatic activity of sPLA2 from Cdt. The interaction with the calcium-binding loop, as shown in Figure 2A, could explain the enzyme inhibition induced by gallic acid. sPLA2 without GA incubation showed a Vmax of A425 nm/min = 0.18/min in Absorbance units (A425 nm) and a Km of 2. 45 mM relative to the NOBA substrate (*n* = 16, and *p* < 0. 05 *) and after incubation, the purified sPLA2:GA complex showed a higher value of both in Vmax and Km, which were, respectively, A425 nm/min = 0.36/min in Absorbance units (A425 nm) and a Km value of 5.73 mM (*n* = 16, and *p* < 0.05 *). These results emphasize the raise in Km value for GA incubated with PLA2, which suggests that the compound decreases the enzyme affinity with the substrate. In addition, the change in Vmax values suggests that the inhibition of GA does not happen in a specific way. Since sPLA2 activity also depends on the N-terminal or short N-terminal segment of the alpha helix, this change in the alpha helix is confirmed by circular dichroism data (Figure 2E), and the docking results reveal that GA exhibits an irreversible inhibition activity against this protein. The sPLA2 purified from Cdt venom (Figure 2G) was used to induce the inflammatory process in all experiments stated above, and the interaction between the protein and GA can be observed in Figure 2F.

## 4. Discussion

There is a consensus that gallic acid could have several beneficial effects. It is a promising candidate for the development of new drugs for experimental use, and there is a call for its use as a nutritional supplement. Other literature data show that GA is a powerful inhibitor of the pharmacological activity of PLA2 from venoms [13,18]. Thus, the major question was what effects of GA would be observed when the compound was previously administered in animals on the development of edema and myotoxicity induced by sPLA2. Our data confirmed the findings revealing that GA, when incubated with the venom PLA2, was able to neutralize the effect of the protein. On the other hand, we also observed that the application of GA before the PLA2 injection neither inhibited the edema nor the myotoxicity induced by the enzyme. Additionally, the experimental data show that the animals treated with GA before and after the injection of PLA2 also showed an increase in the levels of PAF when compared with the control group.

PAF presents a wide range of actions and may strongly contribute to the course of pathophysiologic effects observed during acute inflammation, including the increase of oxidative burst and eicosanoid production [19,20,21]. Previous work showed that Cdt sPLA2 induces a moderate acute edema that involves a rapid mobilization of AA, including the phospholipase C (PLC) protein kinase C (PKC) signaling and the activation of cPLA2, plus a massive production of PAF, which is a biologically active phospholipid provided from the enzymatic action of sPLA2 and cPLA2 [20,21]. Thus, PAF and AA are crucial for the edema and the pathological damage induced by sPLA2 from Cdt [20,22]. PAF degradation requires the acetyl group removal at the sn-2 position of the glycerol backbone, an action catalyzed by PAF-AH, and this is a crucial first step for neutralization of PAF biding to PAF-receptor to inhibit the capacity of the acute inflammatory process by PAF [19,21]. These PAF-AH enzymes are Ca2+-independent which belong to group VII of phospholipase A2. This group of enzymes exists in both extracellular and intracellular forms in various organisms, and the enzyme extracellular PAF-AH is the most important way to control plasma extracellular PAF levels [23,24,25,26]. Thus, the experimental data from GA-treated animals could suggest that the increased plasma level of PAF could be due to inhibition of the extracellular PAF-AH enzyme. The results regarding the interaction between GA and PAF-AH were performed using in silico analysis with modeling tools. Figure 3 reveals that GA strongly interacts with certain crucial amino acid residues found in the extracellular PAF-AH enzyme, residues S248, D271, and H326, which are crucial for its enzymatic activity [24,25]. Furthermore, using the kit (PAF Acetylhydrolase Inhibitor Screening Assay Kit (ab133091), (Abcam Discovery Drive, Cambridge Biomedical Campus, Cambridge, CB2 0AX, UK)) and following the method described by the user manual of the kit, we were able to show that the standard inhibitor (Methyl Arachidonyl Fluorophosphonate, MAFP) showed an IC50 of 250 nM and GA showed an IC50 value of 125 nM; thus GA was twice as effective in inhibiting the standard inhibitor itself. Furthermore, the compound exhibits a high ability to interact at the active site of acetylcholinesterase, binding with S125, H447, and E202 that form the catalytic unit of these enzymes [25]. The sPLA2 from CDT can induce edema through peripheral nerve activation, a process that involves acetylcholine participation. This fact contributes to the neurotoxic, edematogenic, myotoxic, and insulinotropic activities of sPLA2 in an indirect manner [26,27,28,29].

In this article, we observed a significantly increased myotoxicity in animals pre-treated with GA, and this result was somewhat unexpected since the literature strongly suggests that natural compounds would be able to neutralize the myotoxic effect induced by secretory venom PLA2. This property could involve the antioxidant capacity of the compounds as well as their ability to neutralize proteins involved in the inflammatory process, and venom PLA2 itself is also affected by natural compounds [13,20,21,22,23]. Besides that, considering some studies [27], we performed other molecular modeling studies that suggested that GA could inhibit the acetylcholinesterase from peripheral nerve endings, which would be stimulated by sPLA2 from CDT, a potent neurotoxin [29,30,31]. To confirm the inhibitory effect of GA, we used an Acetylcholinesterase Inhibitor Screening Kit Colorimetric (ab283363, Abcam Discovery Drive, Cambridge Biomedical Campus, Cambridge, CB2 0AX, UK). Following the method described by the kit user manual, we were able to show that the standard inhibitor (Donepezil), which comes with the kit, showed an IC50 of 31 nM and GA showed an IC50 value of 125 nM). GA showed a higher value of inhibition than the Donepezil control provided by the kit, but this compound is still a potential inhibitor that can contribute strongly to increased myotoxicity. In addition, these results suggest that PAF could also be an important factor in the establishment of myotoxicity [22,23]. Hence, our experimental data with the support of the in silico data suggest that GA would inhibit, in diverse degrees, the PAF-AH and Acetylcholinesterase. The pharmacological action of PLA2 and other venom toxins also involve the enzymatic mobilization of cPLA2 and increased expression of COX-2. Additionally, more recent data show that the action of COX-2 seems to be crucial for the edema course [20,32,33,34,35,36]. Other studies [34] suggest that GA may also potentially inhibit the COX-2 activity and increase AA levels. This fact may be an important factor for increased oxidative stress, raising the H2O2 and ROS, which can lead to AA peroxidation, membrane destruction, and increased edema, for example [20,21,36].

In addition, PAF can induce edema by other pathways that do not depend on the mobilization of cytosolic PLA2, and that includes, for example, PLC [36,37]. On the other hand, the model usually used for anti-inflammatory assays is based on the lipopolysaccharide (LPS) use as an inflammatory agent, which acts directly on COX-2 and is accompanied by the production and release of pro-inflammatory cytokines, such as interleukin (IL)-1β, IL-6 and Tumor necrosis factor-alpha (TNF-α) [38,39] and which is not the same pathway of inflammation induced by sPLA2 from *Crotalus durissus terrificus*. Thus, the two models are different and produce complementary rather than mutually excluding anti-inflammatory results.

## 5. Conclusions

Gallic acid can be a valuable compound in terms of raw material and it has the ability to act as an anti-inflammatory molecule inhibiting the sPLA2, as an antioxidant agent, and as a COX2 inhibitor [1,13,34,35]. In addition, GA also exhibits other activities such as antimicrobial, anticancer, cardio-protective, gastro-protective, neuroprotective effects [2]. The purpose of this article was not to dispute the studies applied so far; however, GA in experimental conditions injected 30 min before the application of sPLA2 potentiated the inflammatory and myotoxic effects of the PLA2 of rattlesnake. The use of molecular docking tools was crucial to reveal that GA was also able to inhibit PAF-AH and that it induced an increase in PAF concentration that may have played a key role in the inflammatory and myotoxic action of sPLA2. On the other hand, the inhibition of Acetylcholinesterase may have also contributed to the increase in myotoxicity (Figure 3). Thus, another important conclusion of the work is that GA does not inhibit only one enzyme, such as sPLA2, but also several other alpha-beta hydrolases like cholinesterases, PAF-AH, and probably other esterases/lipases. Finally, as stated in the title, GA has two sides or aspects that complement each other, one confirming that GA is a compound with great therapeutic application mainly for inflammation, and the other revealing that the use of this natural compound as a drug without side effects or in a concentrated way should be avoided. More research is necessary for the safe use of GA or other compounds, including the use of sPLA2 as an inflammation-inducing agent next to LPS, for example.

## Figures and Tables

**Figure 1 pharmaceutics-14-00368-f001:**
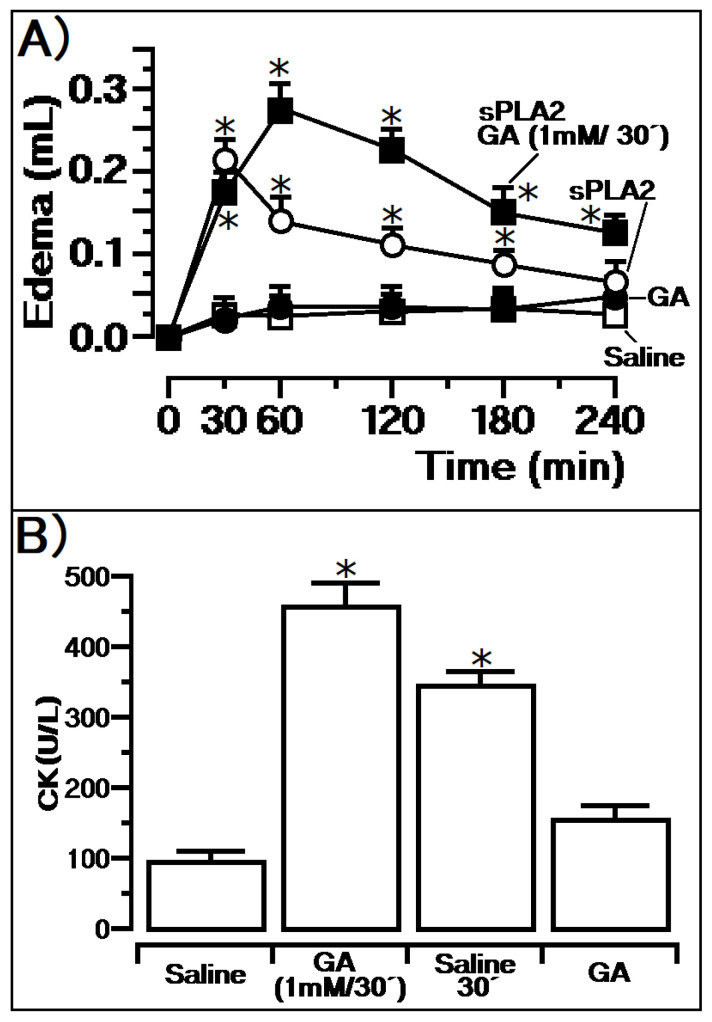
Evaluation of the GA previous treatment on the edema and myotoxicity triggered by sPLA2 from Cdt. In (**A**), the effect of the previous injection of GA, revealing an edema peak at 60 min, which is higher than the positive control (sPLA2) (*n* = 5, and * *p* < 0.05). In (**B**), myotoxic effect results of the animal treated previously with GA(GA (1 mM/30’) reveals a raise in the CK values when compared with the positive and negative controls (*n* = 5, and * *p* < 0.05).

**Figure 2 pharmaceutics-14-00368-f002:**
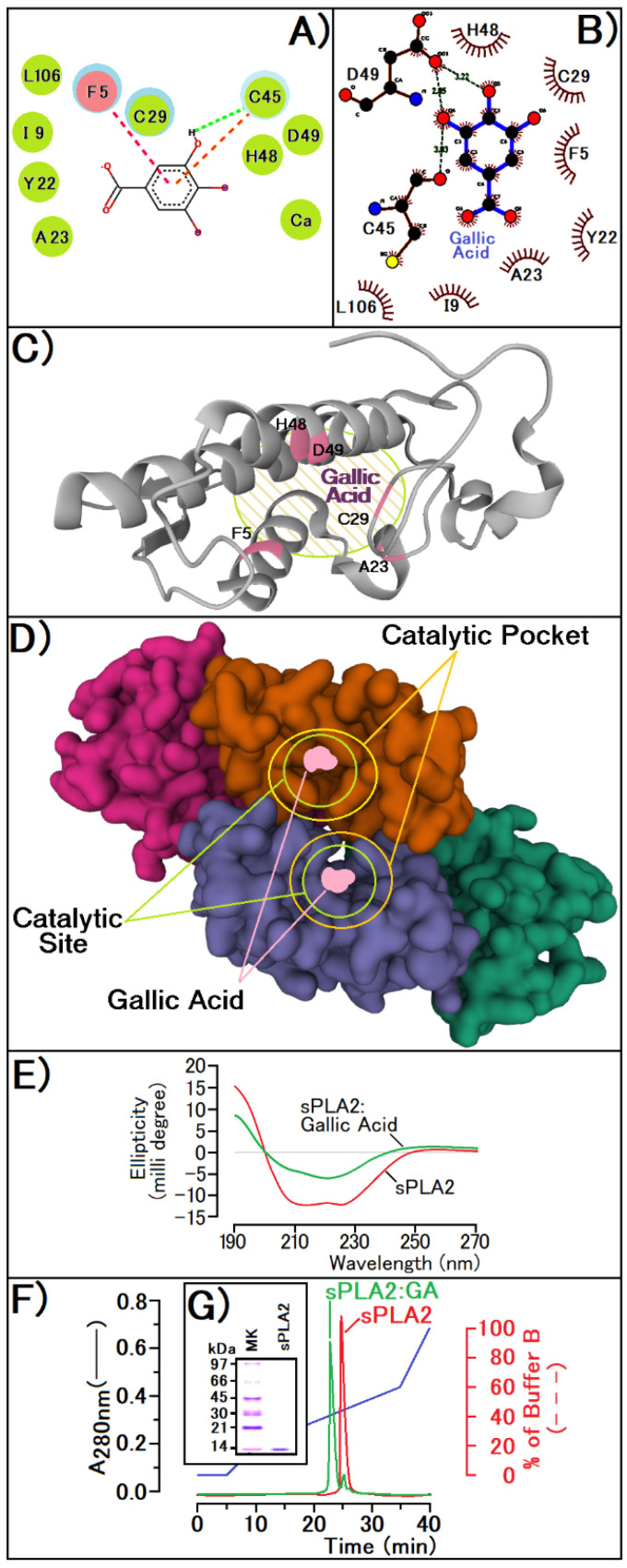
(**A**,**B**) show the molecular interaction models of GA with the active site of PLA2 using Discovery and LigProt programs, respectively. The figures exhibit the main groups of the ligand and proteins responsible for the interaction with the active site of sPLA2 from *Crotalus durissus terrificus*. In (**C**,**D**), it is possible to notice the interaction sites of the compound with the active site of sPLA2 in monomeric and tetrameric form. (**E**) reveals the sPLA2 conformational changes of the secondary structures before and after treatment with GA, using circular dichroism. (**F**) exhibits the reverse phase HPLC profile of sPLA2 before and after modification with GA. (**G**) shows the tricine SDS-PAGE of sPLA2 from *Crotalus durissus terrificus*. Both protocols of the reverse phase HPLC analysis and PAGE SDS Cotrim et al., 2011 [16].

**Figure 3 pharmaceutics-14-00368-f003:**
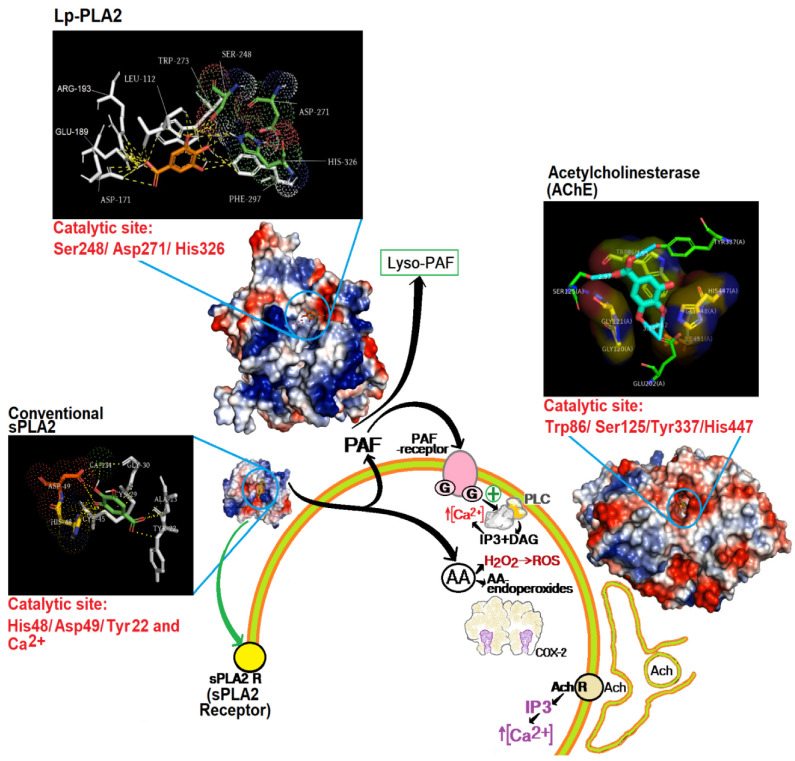
Possible actions of GA that led to increased levels of PAF and AA. The compound interacts with sPLA2 at the active site, binding with essential amino acid residues located in the N-terminal region (F5) and with Histidine (H48) and Aspartic acid (D49). The compound was able to bind with the catalytic site of PLA2 associated with lipoprotein (PAF Acetylhidrolase extracellular), including Ser248 and His 326 and Ser 125 and Tyr337 of the catalytic site of Acetylcholinesterase.

## Data Availability

All data in the paper followed all current analyses and are original.

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
