# Peer review of "Gallic Acid as a Non-Selective Inhibitor of α/β-Hydrolase Fold Enzymes Involved in the Inflammatory Process: The Two Sides of the Same Coin"

_pharmaceutics, 2022, doi:10.3390/pharmaceutics14020368_

Round 1

Reviewer 1 Report

This paper reports data on the effect of gallic acid (GA) on α/β-hydrolase fold enzymes involved in the inflammatory process. Although the manuscript presents some potentially interesting results, there are a number of points concerning their presentation that need to be addressed.

Specific comments

  1. It is recommended to slightly modify the current title of the paper: “Gallic acid as a non-selective inhibitor of α/β-hydrolase: the two sides of the same coin” as follows: “Gallic acid as a non-selective inhibitor of α/β-hydrolase fold enzymes involved in the inflammatory process: the two sides of the same coin”.
  2. Line 213 and line 215 – the units for the given Vmax values (0.18 and 0.36) should be adequately presented (A425nm/min = 0.18/min and A425nm/min = 0.36/min ???).
  3. Lines 210-212 and lines 214-216 – the sentence: “The interaction with the calcium-binding loop, as shown in figure 2A, can explain the enzymatic inhibition by decreasing the Km and Vmax values of the enzyme” and the statement: “after incubation, the purified sPLA2:GA complex reveals a decrease in both Vmax and Km, which were respectively 0.36 in Absorbance units (A425nm) and a Km value of 5.73 mM” require some revision, because according to the data presented in the manuscript, the Km value found for the sPLA2:GA complex is higher compared to the Km value (2.45 mM) found for sPLA2 in the absence of GA.
  4. Figure 2E – the curves showing conformational changes in secondary structures of sPLA2 before and after treatment with GA should be accordingly labeled.
  5. Figure 3 requires some graphical adjustment to make it easier to analyze. Furthermore, the full names for the abbreviations used should be given in the legend.

Minor remarks

  1. Line 135 – it is suggested to replace the term: “The compounds were taken from PubChem platform” by the term: “Information on the structure of gallic acid was taken from PubChem platform”.

Author Response

pharmaceutics-1523995

Reviewer 1:

All modifications are highlighted in yellow.

Question1: It is recommended to slightly modify the current title of the paper: “Gallic acid as a non-selective inhibitor of α/β-hydrolase: the two sides of the same coin” as follows: “Gallic acid as a non-selective inhibitor of α/β-hydrolase fold enzymes involved in the inflammatory process: the two sides of the same coin”.

Answer: OK, we changed the title according to the reviewer's suggestion.

Question 2: Line 213 and line 215 – the units for the given Vmax values (0.18 and 0.36) should be adequately presented (A425nm/min = 0.18/min and A425nm/min = 0.36/min ???).

Answer: We appreciate your comment, and the modifications were already done.

Question3: Lines 210-212 and lines 214-216 – the sentence: “The interaction with the calcium-binding loop, as shown in figure 2A, can explain the enzymatic inhibition by decreasing the Km and Vmax values of the enzyme” and the statement: “after incubation, the purified sPLA2:GA complex reveals a decrease in both Vmax and Km, which were respectively 0.36 in Absorbance units (A425nm) and a Km value of 5.73 mM” require some revision, because according to the data presented in the manuscript, the Km value found for the sPLA2:GA complex is higher compared to the Km value (2.45 mM) found for sPLA2 in the absence of GA.

We rewrite the sentence to explain these results in this part: “Thus, in 2C we present a possible interaction site of GA with sPLA2, exhibiting the involved residues that are crucial for the enzymatic activity of sPLA2 from Cdt. The interaction with the calcium binding loop, as shown in Figure 2A, could explain the enzyme inhibition induced by gallic acid. sPLA2 without GA incubation showed a Vmax of A425nm/min = 0.18/min in Absorbance units (A425nm) and a Km of 2. 45 mM relative to the NOBA substrate (n=16, and P<0. 05*) and after incubation, the purified sPLA2:GA complex showed a higher value of both in Vmax and Km, which were respectively A425nm/min = 0.36/min in Absorbance units (A425nm) and a Km value of 5.73 mM (n=16, and P<0.05*).   Thus, the increase in Km value for GA incubated with PLA2 suggests that the compound decreases the affinity of the enzyme with the substrate. Besides, the change in Vmax values suggest, that the inhibition of GA does not happen in a specific way.  Since sPLA2 activity also depends on the N-terminal or short N-terminal segment of the alpha helix, this change in the alpha helix confirmed by circular dichroism data (Figure 2E) and the docking results reveal that GA exhibits an irreversible inhibition activity against this protein.” 

Question 4: Figure 2E – the curves showing conformational changes in secondary structures of sPLA2 before and after treatment with GA should be accordingly labeled.

Answer: The figure has been modified and the annotation requested by reviewer 1 has been added in figure 2E.

Question5: Figure 3 requires some graphical adjustment to make it easier to analyze. Furthermore, the full names for the abbreviations used should be given in the legend.

Answer: The names of the enzymes have been placed in the figure, which has also been resized to try to improve the interpretation of the figure.  We also put two more pieces of information in the legend to highlight some important points in figure 3.

Minor remarks

Line 135 – it is suggested to replace the term: “The compounds were taken from PubChem platform” by the term: “Information on the structure of gallic acid was taken from PubChem platform”.
Answer: The sentence was modified. 

Reviewer 2 Report

Abstract 

  • There is no consistent connection between the title and the abstract. Rewrite the abstract or the title.

Introduction 

  • Unclear research question. 

Results 

  • There is no result regarding the purification of gallic acid. How pure it was it obtained? Also, a discussion on the effect of impurities is needed. 
  • Lacking on the purification of sPLA2. HPLC results are not shown in the manuscript. Standard SDS-PAGE analysis is also required. 
  • The correlation between the experimental data and docking studies must be based on the experimental determination of the kinetic constants, which are lacking in this work. 

Discussion 

  • The connection between the experimental and theoretical parts should need more elaboration.

Conclusion 

  • There is no clear research question, not enough experimental support, and weekly discussions cannot support the conclusion.

Author Response

pharmaceutics-1523995

Reviewer 2

Question 1: Abstract 

  • There is no consistent connection between the title and the abstract. Rewrite the abstract or the title.

Answer: Firstly, we appreciate all comments, suggestions, and questions. We changed the abstract and the title to improve the connection.

Original Abstract:

(1)       Background: Gallic acid (GA) is a natural phenolic compound that has been described as a promising candidate for new drug discovery as well as a potential apoptosis-inducing agent. (2/3) Methods and Results: GA was isolated from the ethanolic extract of fresh pitanga (Eugenia uniflora) leaves, which neutralized the enzymatic activity of the secretory PLA2 (sPLA2) by inhibiting the active site and inducing changes in the secondary structure of the enzyme. Pharmacological assays of edema showed that GA strongly decreased edema when the compound was previously incubated with sPLA2. However, a previous GA treatment (30 minutes before) significant increase the edema and myotoxicity induced by sPLA2. Molecular docking results of GA with platelet-activating factor-acetylhydrolase (PAF-AH), acetylcho-linesterase and cytosolic (cPLA2) showed that this compound interacts strongly with PAF-AH and acetylcholinesterase and has no interaction with the active site of cytosolic PLA2. (4) Conclusion: sPLA2 is an inflammatory agent that induces the production of platelet-activating factor (PAF), arachidonic acid (AA) and a neuromuscular facilitator toxin. Thus, GA can inhibit PAF-AH, which allows free action of PAF increasing edema. Besides, PAF-AH inhibition potentially blocks acetylcholinesterase and with PAF may have increased creatine kinase (CK) release into the blood.

New Version of Abstract.

(1) Background: Gallic acid (GA) has been characterized as an effective anti-inflammatory, antivenom and a promising drug for therapeutic use. (2/3) Methods and Results: GA was identified from ethyl extract of fresh pitanga (Eugenia uniflora) leaves which was identified using commercial GA.  Commercial GA neutralized the enzymatic activity of secretory PLA2 (sPLA2) by inhibiting the active site and inducing changes in the secondary structure of the enzyme.  Pharmacological edema assays showed that GA strongly decreased edema when the compound was previously incubated with sPLA2. However, a prior treatment of GA (30 minutes before) significantly increased the edema and myotoxicity induced by sPLA2.  The molecular docking results of GA with platelet-acetylhydrolase (PAF-AH) and acetylcholinesterase reveal that this compound was able to interact with the active site of both molecules, inhibiting the hydrolysis of platelet activating factor (PAF) and acetylcholine (ACh). (4) Conclusion: GA has a great potential application however, our results show that GA can also induce adverse effects in previously treated animals. Additionally, the increased edema and myotoxicity observed experimentally in GA treated animals may be due to the inhibition of PAF-AH and Acetylcholinesterase.

Question 2: Introduction 

  • Unclear research question. 

Answer: The research question was added in the introduction and discussion.

Question 3: Results 

  • There is no result regarding the purification of gallic acid
  • How pure it was it obtained? Also, a discussion on the effect of impurities is needed. 

Answer: We have already corrected these points along the text. In this way, we explained in the results that we used for the GA identification in the samples a standard GA commercially purchased from Sigma and that this compound was used to perform the experiments. Thus, if there are impurities in the gallic acid samples we do not know, because we rely on the data provided by the company.

In text, we performed this modification:In this work we used commercial gallic acid that allowed identification in the pre-pared pitanga leaf samples based on the methods described by Assunção et al, 2017 [15], including sample preparation using ultrasound-assisted extraction followed by a concen-tration. The material was cleaned using solid phase extraction cartridges prior to fraction-ation on reversed phase HPLC using a Supelco Analytical C18 column (250 × 4. 6 mm, 5 μm); mobile phase water (A)/acetonitrile (B). GA identification was confirmed by com-mercial gallic acid marker (Gallic acid monohydrate, ACS 98%, Sigma-Aldrich) and the respective UV scanning of both compounds from sample and standard.

Question4: Lacking on the purification of sPLA2. HPLC results are not shown in the manuscript. Standard SDS-PAGE analysis is also required. 

Answer: Figure 2F and 2G has been changed to fit the requests of the second reviewer.

Question 5: The correlation between the experimental data and docking studies must be based on the experimental determination of the kinetic constants, which are lacking in this work. 

Answer: We have included two experimental results to try to remedy some of the problems in the article and have placed them in the discussion part (highlighted).  We did not put the results of the cPLA2 activity because of failure in the reagent supply chain and because of various import problems, and so we removed cPLA2 from figure 3.

Question 6: Conclusion 

  • There is no clear research question, not enough experimental support, and weekly discussions cannot support the conclusion.

Answers: the modification in all these parts were made and the conclusion was corrected:

Conclusions

Gallic acid can be a valuable compound in terms of raw material and with the ability to act as an anti-inflammatory molecule inhibiting the sPLA2, as an antioxidant agent and as a COX2 inhibitor [1, 13, 34, 35/ 1, 13, 35, 36]. In addition, GA also exhibits other activities such as antimicrobial, anticancer, cardio-protective, gastro-protective, neuroprotective effects [2]. The purpose of this article was not to dispute the studies applied so far, however, GA in experimental conditions injected 30 minutes before the application of PLA2 potentiated the inflammatory and myotoxic effects of the PLA2 of rattlesnake.  The use of molecular docking tools were crucial to reveal that GA was also able to inhibit  PAF AH and that it induced an increase in PAF concentration that may have played a key role in the inflammatory and myotoxic action of PLA2. On the other hand, the inhibition of Acetylcholinesterase may have also contributed to the increase in myotoxicity (Figure 3).  Thus, another important conclusion of the work is that GA does not inhibit only one enzyme suck as sPLA2, but also several other alpha-betha hydrolases like cholinesterases, PAF-AH and probably other esterases/ lipases. Finally, as stated in the title, GA has two sides or aspects that complement each other: one confirming that GA is a compound with great therapeutic application mainly for inflammation, and other revealing that the natural compound should not be used as a drug without side effects and do not should be used in a concentrated way. It is necessary more research for the safe use of GA or other compounds including the use of secretory PLA2 as an inflammation-inducing agent next to LPS for example.

Round 2

Reviewer 2 Report

The authors have corrected and clarified the issues pointed out previously. The current version of the manuscript shows improvements.